# Identification of Novel Senescent Markers in Small Extracellular Vesicles

**DOI:** 10.3390/ijms24032421

**Published:** 2023-01-26

**Authors:** Tomoka Misawa, Kazuhiro Hitomi, Kenichi Miyata, Yoko Tanaka, Risa Fujii, Masatomo Chiba, Tze Mun Loo, Aki Hanyu, Hiroko Kawasaki, Hisaya Kato, Yoshiro Maezawa, Koutaro Yokote, Asako J. Nakamura, Koji Ueda, Nobuo Yaegashi, Akiko Takahashi

**Affiliations:** 1Division of Cellular Senescence, The Cancer Institute, Japanese Foundation for Cancer Research, Tokyo 135-8550, Japan; 2Department of Obstetrics and Gynecology, Tohoku University Graduate School of Medicine, Miyagi 980-8575, Japan; 3Graduate School of Science and Engineering, Ibaraki University, Ibaraki 310-8512, Japan; 4Project for Personalized Cancer Medicine, Cancer Precision Medicine Center, Japanese Foundation for Cancer Research, Tokyo 135-8550, Japan; 5Department of Endocrinology, Hematology and Gerontology, Graduate School of Medicine, Chiba University, Chiba 260-0856, Japan; 6Advanced Research & Development Programs for Medical Innovation (PRIME), Japan Agency for Medical Research and Development (AMED), Tokyo 104-0004, Japan; 7Cancer Cell Communication Project, NEXT-Ganken Program, Japanese Foundation for Cancer Research, Tokyo 135-8550, Japan

**Keywords:** biomarker, cellular senescence, EV proteomics, SASP, Werner syndrome

## Abstract

Senescent cells exhibit several typical features, including the senescence-associated secretory phenotype (SASP), promoting the secretion of various inflammatory proteins and small extracellular vesicles (EVs). SASP factors cause chronic inflammation, leading to age-related diseases. Recently, therapeutic strategies targeting senescent cells, known as senolytics, have gained attention; however, noninvasive methods to detect senescent cells in living organisms have not been established. Therefore, the goal of this study was to identify novel senescent markers using small EVs (sEVs). sEVs were isolated from young and senescent fibroblasts using three different methods, including size-exclusion chromatography, affinity column for phosphatidylserine, and immunoprecipitation using antibodies against tetraspanin proteins, followed by mass spectrometry. Principal component analysis revealed that the protein composition of sEVs released from senescent cells was significantly different from that of young cells. Importantly, we identified ATP6V0D1 and RTN4 as novel markers that are frequently upregulated in sEVs from senescent and progeria cells derived from patients with Werner syndrome. Furthermore, these two proteins were significantly enriched in sEVs from the serum of aged mice. This study supports the potential use of senescent markers from sEVs to detect the presence of senescent cells in vivo.

## 1. Introduction

Functional decline and impairment accompanied by aging have become serious problems worldwide. Aging in mammals is characterized by several cellular and molecular events [1]. Cellular senescence is one hallmark of aging that is a consequence of the stress response. The resulting inhibition of cell proliferation has both physiological and pathological aspects [2]. The accumulation of senescent cells during the aging process is associated with some typical features, such as the increased expression of cyclin-dependent kinase inhibitors, p16^INK4a^ and p21^WAF1/CIP1^, and β-galactosidase activity [3,4]. Moreover, a secretory phenomenon of proinflammatory proteins, known as the senescence-associated secretory phenotype (SASP), has an important role in the pathophysiological function of senescent cells because of its deleterious effects associated with age-related diseases, including cancer, dementia, chronic obstructive pulmonary disease, and osteoporosis [5]. In addition to inflammatory proteins, senescent cells secrete small extracellular vesicles (EVs), which enhance the proliferation of cancer cells, paracrine senescence, and chromosomal instability that acts as one of the SASP factors [6,7,8,9].

EVs and nanoparticles are released from all types of mammalian cells. They consist of various cell components and transfer information to surrounding tissues [10]. In particular, small EVs (sEVs), including exosomes, exhibit a variety of biological functions depending on in vivo conditions [11,12]. Because sEV secretion is activated in senescent cells, a significant increase in the number of EVs occurs in body fluids [13,14,15,16,17]. Profiling sEVs from liquid biopsy samples may result in the identification of a senescence signature that could be useful for clinical diagnosis and therapeutic intervention with senolytics [18,19,20,21]. Indeed, some attempts have been made to validate a senescent signature for sEVs in the context of aging biomarker development [14,22,23,24,25]. Most of these reports used non-specific polymer-based precipitation or ultracentrifugation techniques to collect sEVs samples; however, recent studies indicate that there is heterogeneity among sEVs, and the non-vesicular extracellular matter is often detected when using these methods [10]. Therefore, we purified sEVs using three methods available in clinical settings and conducted a mass spectrometry (MS) analysis to identify new biomarkers for sEVs. In addition, we confirmed the presence of sEVs derived from patients with Werner syndrome and serum from aged mice.

## 2. Results

### 2.1. Small EVs Collected by Three Different Methods

To identify proteins that are more abundant in senescent cell-derived EVs compared with young cell-derived EVs, sEVs were isolated from cell culture supernatants using three methods that are capable of collecting highly pure sEVs. Cellular senescence was induced using serial passage (replicative senescence: RS) or X-irradiation (IR) in normal human diploid fibroblasts (TIG-3 cells) (Figure 1A). Senescence induction was assessed via the expression level of p16^INK4a^, p21^WAF1/CIP1^, SASP factors (Interleukin-6: IL-6 and IL-8), decreased lamin B1 levels, the accumulation of DNA damage foci, and Senescence-associated β-gal staining (Figure 1B–D). Young control (Ctrl) and two types of senescent cells (RS and IR) were incubated in a conditioned medium (CM) with 5% EV-depleted FBS for 48 h (Figure 1A). The supernatants were then collected for MS analysis after conducting the sEVs purification procedures (Figure 1E). Since non-EV proteins were also detected in the samples isolated using conventional ultracentrifugation, as previously reported [10], we selected three widely used methods for EV purification. For size-exclusion chromatography (SEC), nanoparticle tracking analysis (NTA) revealed high concentrations of sEV-sized particles around fraction 5. Then, fractions 4–6 were pooled and used as the sEV sample (Appendix A). NTA indicated that the average diameter of the EVs was 141 ± 35 nm (Ctrl-sEV), 153 ± 51 nm (RS-sEV), and 139 ± 43 nm (IR-sEV), with no significant difference in size (Appendix A). The second method involved purification using phosphatidylserine (PS) affinity columns. The presence of tetraspanin proteins (CD9 and CD81), frequently used EV markers, and a non-EV marker protein (calnexin) was confirmed via Western blot analysis (Appendix A). NTA data indicated that the average diameter of the EVs was 143 ± 31 nm (Ctrl-sEV), 151 ± 41 nm (RS-sEV), and 141 ± 38 nm (IR-sEV) and that sEV secretion increased significantly in senescent cells compared with proliferating cells (Appendix A) as previously reported [13]. The third method was immunoprecipitation (IP) using antibodies against three major tetraspanins (CD9, CD63, and CD81). The presence of exosomes in the precipitated samples was confirmed via Western blot analysis (Appendix A). The results indicated that all three purification methods yielded sEVs from the cell culture medium.

### 2.2. Senescent Proteomic Profiles Are Different Depending on the Collection Procedure

To analyze the differences in EV purification procedures, we conducted principal component analysis (PCA) of the 2030 proteins detected by proteomic analysis under nine conditions (Figure 2A). PCA indicated that EV purification methods largely affected their proteomic profiles. In addition, cellular senescence caused a global change in sEV proteins regardless of the method used to induce cellular senescence (Figure 2B and Appendix A). Although PC1 or PC2 values of both SEC and PS collection procedures were very similar, the value of IP was different. Similarly, unsupervised hierarchical clustering analysis also indicated that the IP collection procedure was classified into a different cluster from the SEC or PS collection procedures. Both senescent conditions (RS or IR) were classified into the same cluster for each collection procedure (Figure 2C). To clarify the features of senescent cell-derived EVs, we performed pathway and process enrichment analysis using proteins detected in only the senescent cell-derived EVs in each collection procedure (Appendix A). The analysis revealed various pathways, including those of cellular responses to stress (R-HSA-2262752) and those of membrane trafficking (R-HSA-199991), which were enriched in senescent cell-derived EVs (RS or IR). This indicates that the proteins associated with senescence and secretion of EVs are enriched in both the senescent cell-derived EVs for all collection procedures. Interestingly, for the SEC and PS collection procedures, the pathways of aging (GO:0007568), regulation of I-kappaB kinase/NF-kappaB signaling (GO:0043122), and cytokine signaling in the immune system (R-HSA-1280215) were enriched in senescent cell-derived EVs, which are known to be involved in SASP. This is consistent with previous reports [25]. Altogether, these results indicate that senescent proteomic profiles differ depending on the collection procedure. In addition, senescent cells may secrete inflammatory cytokines via sEVs.

Since collection methods can affect the yield and purity of sEVs, we confirmed the proteomic characteristics by MS using sEVs collected by the three methods (Appendix A). According to the MISEV guidelines, we categorized the detected proteins into two groups: frequently used exosomal markers and non-exosomal markers (Appendix A) [26]. The result indicated that the representative exosomal marker proteins, CD9, CD63, CD81, FLOT1, FLOT2, HSP90AA1, HSP90AB1, PDCD6IP (also known as ALIX), and TSG101 were abundant in sEVs regardless of the collection method (Appendix A). On the other hand, nuclear proteins (NOLC1, SKP1), Golgi proteins (GM130), and mitochondrial proteins (CYC1, COX5B, TOM20) were consistently undetectable in all the samples. Calnexin used as an exclusion marker [26] was also completely undetected in the sEVs collected by IP but was detected at extremely low levels in the sEVs collected by PS and SEC (Appendix A). These results indicate that the sEVs collected using the three methods contained exosome fractions as defined by the MISEV guidelines.

Using the results of our MS analysis, we identified thirteen proteins with an abundance ratio of RS-EVs to Ctrl-EVs and an abundance ratio of IR-EVs to Ctrl-EVs that were more than twice in each collection method: ATP6V0A1, ATP6V0D1, CXADR, DNASE1L1, KCTD12, LNPEP, PDIA3, PPIB, P4HB, RTN4, SLC9A1, STX1B, and TTC7B (Figure 2D). We hypothesized that these thirteen proteins might act as senescent EV markers.

### 2.3. Characterization of Small EVs from Patient Cells with Werner Syndrome

To confirm the expression of the thirteen candidates as senescent EV markers, we used the fibroblasts of two patients with Werner syndrome (WF1 and WF2: Werner fibroblasts) and age-matched healthy volunteers (normal fibroblasts: NF), as previously described [27]. We isolated sEVs using SEC and conducted proteomic analysis (Appendix A). To establish differential expression profiles of sEVs from NF and WF, we performed a cluster analysis using the proteomics data. The results indicated that NF and WF exhibit different proteomic profiles for sEVs. The characteristics of the sEVs from NF after serial passage (PDL42) were similar to those from WF compared with non-passage NF (PDL22) (Figure 3A). Moreover, Gene Ontology (GO) analysis of the WF data indicated that the component proteins of the EVs were enriched in protein binding, calcium-dependence, cadherin binding involved in cell–cell adhesion, proton-transporting ATPase activity, and rotational mechanisms (Figure 3B). Upon comparison of the proteomic profiles between highly detected EV proteins in WF and thirteen putative senescent EV markers obtained from Figure 2D, we identified two proteins in common, ATP6V0D1 and RTN4 (Figure 3C). Consistent with the increase in some senescent markers (p16^INK4a^, p21^WAF1/CIP1^, and IL-6), we found that the mRNA expression levels of ATP6V0D1 and RTN4 were also upregulated in WF (Figure 3D). Therefore, we focused on investigating ATP6V0D1 and RTN4 as novel markers for sEVs obtained from senescent cells.

### 2.4. Two Proteins Are Enriched in Small EVs from Senescent Cells, Patient Cells with Progeria, and Serum from Aged Mice

To verify their utility as senescence markers in sEVs, we measured the expression level of the two candidate senescence markers in TIG-3 cell-derived EVs and in WF-derived EVs. In TIG-3 cells, the expression of ATP6V0D1 and RTN4 in whole-cell lysates (WCL) was high in RS and IR cells compared with the control cells (Figure 4A). In addition, those proteins were significantly enriched in sEVs released from senescent cells (RS and IR) (Figure 4B). Moreover, ATP6V0D1 and RTN4 were highly expressed in cells of the patient with Werner syndrome (WF1 and WF2) compared with cells derived from healthy volunteers (NF) in WCL and were also highly detectable in sEVs from WF (Figure 4C,D).

Finally, to determine the efficacy of the two candidate proteins as senescence markers using sEVs during the aging process, we conducted a proteomic analysis using serum-derived EVs from young (8 weeks age) and aged mice (110 weeks age) (Appendix A). The results indicated that both ATP6V0D1 and RTN4 were significantly enriched in sEVs from the serum of aged mice (Figure 4E). Therefore, we conclude that these two proteins in sEVs may serve as indicators of the presence of senescent cells in vivo.

## 3. Discussion

The secretion of sEVs is upregulated in senescent cells compared with young cells through the activation of the p53 pathway [28,29] or the ceramide pathway [13,15]. In the present study, it was found that the protein expression profiles of sEVs released from normal and senescent cells were quite different and that the isolation procedures must be considered when analyzing the proteomic profiles of sEVs. Recent studies have identified the alterations of the protein, miRNA, and lipid composition of sEVs during cellular senescence, which result in significant changes in biological function [6,22,24,30]. However, these studies prepared sEVs using conventional methods, which may have introduced non-vesicle proteins. The present study is the first to show differences in the profiles of sEVs derived from senescent and young cells isolated by three different methods. However, the mass spectrometric analysis performed in this study cannot completely exclude the possibility of including noise and positive/negative failure. The complexity and heterogeneity of cellular senescence are both highly dependent on stressors, cell types, and physiologic conditions [5,24,31]. The lack of a sensitive and specific marker to assess cellular senescence aggravates this complexity. For example, SASP factors are not specific to senescent cells; they are also often secreted by non-senescent cells, such as tumor cells or immune cells, which makes it difficult to evaluate cellular senescence by SASP factors alone [5]. It is eventually expected that SASP factors will be detectable from a blood sample. However, since blood contains albumin, many free proteins and protein aggregates, methods of detecting these proteins in blood components may not be sensitive enough. In contrast, proteins within sEVs are stable because they possess a lipid bilayer structure. Therefore, proteins enriched in sEVs (ATP6V0D1 and RTN4) may increase the sensitivity and accuracy of the detection of senescent cells from blood samples.

The presence of senescent cells in the tissue microenvironment promotes tumorigenesis, fibrosis, and inflammation; these result in age-related diseases, such as cancer [17,32,33,34]. sEVs released from senescent cells also participate in various biological functions, such as cancer cell proliferation [6] and malignant transformation [35], the induction of chromosome instability [9], and the propagation of cellular senescence [7,14]. Senolytic treatments, which selectively induce cell death and the elimination of senescent cells, have been developed over the last decade [36,37]. In addition, senolytic drugs (drugs that target senescent cells) may be used in combination with anticancer agents and other drugs because cellular senescence is not only caused by aging but also occurs in cancer cells following treatment with anticancer agents and other therapeutics [38]. However, there are currently no established, noninvasive methods to detect senescent cells in vivo; such methods would allow the accurate determination of the timing of senolytic drug administration or the evaluation of effectiveness in removing senescent cells from the body. Therefore, we identified senescent EV markers as means to detect senescent cells from samples of bodily fluids such as blood. It may also be possible to combine senolytic drugs with anticancer drugs and other therapies, thereby reducing the incidence of age-related diseases, including cancer, and extending life expectancy.

Furthermore, we confirmed the correlation between the gene expression levels of thirteen candidates of senescent markers detected in this study with human age using previously reported RNA sequence data from human dermal fibroblasts collected from individuals (between 1 and 94 years) of age (GSE113957, [39]) (Appendix A). First, we analyzed the gene expression levels of conventional senescence markers (p21^WAF1/CIP1^ and IL-6) in clinical samples from young and old humans. The expression of p21^WAF1/CIP1^ and IL-6, a SASP factor, was positively correlated with chronological age (Appendix A). As expected, the expression of lamin B1 was negatively correlated with age (Appendix A). Of the thirteen candidate genes, three (ATP6V0A1, KCTD12, and RTN4) were significantly upregulated in older people (Appendix A). The predictive ability of chronological age evaluated using ATP6V0A1, KCTD12, or RTN4 levels was found to be equivalent to that of biological age (ATP6V0A1 ROC AUC = 0.7; KCTD12 ROC AUC = 0.7; RTN4 ROC AUC = 0.78) (Appendix A). Logistic regression analysis indicated that ATP6V0A1, KCTD12, and RTN4 were the strongest biomarkers associated with chronological age (ATP6V0A1, KCTD12, and RTN4 ROC AUC = 0.83) and were potentially greater than the predictive value of the major senescence markers (p21^WAF1/CIP1^ ROC AUC = 0.75; p16^INK4a^ ROC AUC = 0.56; IL-6 ROC AUC = 0.61) (Appendix A). Taken together, our data suggest that the gene expression levels of ATP6V0A1, KCTD12, and RTN4 are useful for the prediction of senescent cells in the human body.

From the proteomic analysis using sEVs, we identified ATP6V0D1 and RTN4 as novel markers to detect senescent cells in vivo (Figure 5). RTN4, a member of the reticulon protein family, is localized within the ER membrane [40]. In previous studies, the overexpression of RTN4 in prostate cancer cells was shown to block the cell cycle in the G2/M phase and to induce cell senescence [41] and the expression of inflammatory cytokines, such as CHOP, IL-6, and tumor necrosis factor (TNF)-α, in a muscle cell line (C2C12) [42]. Thus, RTN4 appears to be associated with the induction of senescence. In contrast, ATP6V0D1, a subunit of the transmembrane V0 complex of the proton-transporting V-type ATPase, which is located on the lysosomal membrane [43], has not been previously associated with aging. It is not yet clear why ATP6V0D1 and RTN4 proteins are highly enriched in sEVs in aged samples; therefore, further investigation is warranted. We have presented the first demonstration that these two proteins are strong candidates for senescence markers using sEVs. In the future, it is expected that the detection of these senescence markers in human-serum-derived EVs will facilitate the properly managed administration of senolytic drugs for the prevention or treatment of age-related diseases.

## 4. Materials and Methods

### 4.1. Cell Culture

TIG-3 cells were obtained from the Japanese Collection of Research Bioresources and cultured in Dulbecco’s Modified Eagle’s Medium (DMEM) (Nacalai Tesque, Kyoto, Japan) supplemented with 10% fetal bovine serum (FBS) and 1% penicillin/streptomycin (Sigma-Aldrich, MO, USA). Early passage TIG-3 cells (<40 population doublings) were considered young cells (control), and late passage TIG-3 cells (>70 population doublings) that had ceased to proliferate were considered replicative senescent cells. For senescence induction, the cells were irradiated with 10 Gy X-rays using a CP-160 model (Faxitron X-ray, Inc., AZ, USA). Young and senescent cells were cultured in a 5% CO_2_ incubator maintained at 37 °C. The absence of mycoplasma contamination was confirmed in the cultured cells.

### 4.2. Small EV Isolation from Cells

For small EV (sEV) purification, FBS was ultracentrifuged at 100,000× *g* for 16 h to remove microvesicles, and a conditioned medium (CM) was prepared by adding 5% FBS to DMEM. The sEVs were obtained from the supernatant by using the previously described procedure with some modifications [13]. Briefly, the cells were incubated in CM for 48 h. The supernatants were then collected and centrifuged at 300× *g* for 5 min to eliminate the cells, followed by 2000× *g* for 10 min to remove cell debris. Further, the supernatant was centrifuged at 10,000× *g* for 30 min, followed by filtration through a 0.2-µm pore filter (Sartorius Stedim Biotech, Göttingen, Germany) to remove contaminating apoptotic bodies, shed vesicles, and cell debris. The resulting supernatant was used for sEV isolation using the MagCapture Exosome Isolation Kit PS (PS, Fujifilm Wako Chemicals, Tokyo, Japan), size exclusion chromatography via maxiPURE-EVs (SEC, HansaBioMed Life Sciences Ltd., Tallinn, Estonia), or immunoprecipitation with anti-CD9, -CD81, -CD63 antibodies using the ExoCap^TM^ Streptavidin kit (IP, Medical & Biological Laboratories, Tokyo, Japan). sEVs were collected according to the specific manufacturer’s protocol. The number and size of the particles were determined via nanoparticle tracking analysis using a NanoSight LM10 system (Malvern Panalytical Ltd., WR, Germany). These samples were stored at 4 °C or −80 °C until analysis.

### 4.3. Isolation of Small EVs from Mouse Serum

Male C57BL/6 mice (7 weeks old; CLEA Japan Inc., Tokyo, Japan) were maintained under specific pathogen-free conditions. Whole blood was collected from the left ventricle of 8-week-old (n = 5) and 110-week-old (n = 5) mice using a 26G1/2 needle. The blood was allowed to clot for 2 h at room temperature, centrifuged at 1200× *g* for 20 min at 4 °C, and the supernatant was centrifuged at 1200× *g* for 10 min at 4 °C. Further, the supernatant was centrifuged at 10,000× *g* for 30 min at 4 °C, and its resultant supernatant was used as a serum. Approximately 500 µL of serum was obtained from each sample. These samples were stored at −80 °C until analysis. The sEV purification from the mouse serum was conducted using the MagCapture Exosome Isolation Kit PS. The number and size of the particles were measured via nanoparticle tracking analysis using ZetaView (Particle Metrix GmbH Inc., Bavaria, Germany). All of the animal procedures were performed using protocols approved by the JFCR Animal Care and Use Committee. The relevant guidelines and regulations were followed.

### 4.4. Fluorescence Microscopic Analysis

The cells were fixed with 4% paraformaldehyde/PBS (Fujifilm Wako Chemicals, Tokyo, Japan) for 10 min, and membrane permeation was performed using 0.2% Triton X-100/Tris-buffered saline (TBS) for 5 min at room temperature. Next, the cells were blocked with 1% bovine serum albumin and 10% goat serum in TBS for 1 h at 4 °C, then incubated with the following antibodies in the blocking buffer overnight at 4 °C: γ-H2AX (Merck Millipore, Darmstadt, Germany, #05-636, 1:1000 dilution) and phospho-(Ser/Thr) ATM/ATR substrate (Cell Signaling Technology, Danvers, MA, USA, #2851, 1:500 dilution). The secondary antibodies were Alexa488 anti-rabbit IgG and Alexa594 anti-mouse IgG antibodies (Thermo Fisher Scientific, Waltham, MA, USA, 1:1000 dilution) diluted in blocking buffer, and the cells were incubated for 45 min at room temperature. The nucleus was stained with DAPI (Dojindo, Tokyo, Japan) for 5 min at room temperature. After immunostaining, DNA damage-positive cells were quantified by observation under a fluorescence microscope (Carl Zeiss, Oberkochen, Germany).

### 4.5. β-Galactosidase (β-Gal) Staining

The cells were washed in PBS, fixed for 5 min (room temperature) in 2% formaldehyde/0.2% glutaraldehyde, washed, and incubated for 16 h at 37 °C (no CO_2_) with fresh senescence-associated β-Gal (SA-β-Gal) stain solution: 20 mg/mL of 5-bromo-4-chloro-3-indoly β-D-galactoside (X-Gal) in dimethylformamide/200 mM citric acid/sodium phosphate, pH 6.0/100 mM potassium ferrocyanide/100 mM potassium ferricyanide/5 M sodium chloride/1 M magnesium chloride. After staining, SA-β-Gal positive cells were quantified by observation under a fluorescence microscope (Carl Zeiss, Oberkochen, Germany) with a 40× objective lens.

### 4.6. Mass Spectrometry

sEVs were isolated from the CM using three methods and subsequently reduced in a 1× Laemmli sample buffer with 10 mM TCEP, incubated at 100 °C for 10 min, alkylated with 50 mM iodoacetamide at ambient temperature for 45 min, and subjected to SDS-PAGE. Electrophoresis was halted after a migration distance of 2 mm from the top edge of the separation gel. After Coomassie Brilliant Blue staining, the protein bands were excised, destained, and precisely cut prior to in-gel digestion by subjecting it to Trypsin/Lys-C Mix (Promega, WI, USA) at 37 °C for 12 h. The resulting peptides were extracted from the gel fragments and analyzed using an Orbitrap Fusion Lumos Mass Spectrometer (Thermo Fisher Scientific, Waltham, MA, USA) combined with an UltiMate 3000 RSLC nano-flow HPLC (Thermo Fisher Scientific, Waltham, MA, USA). Tandem MS spectra were investigated against a Homo sapiens and Mus musculus protein sequence database in SwissProt using Proteome Discoverer 2.2 (Thermo Fisher Scientific, Waltham, MA, USA), and the peptide identification filters were set to a false discovery rate (FDR) of <1%. Gene ontology analysis was performed using Metascape.

### 4.7. Quantitative Real-Time PCR

The total RNA was extracted from the cultured cells using the mirVana kit (Thermo Fisher Scientific, Waltham, MA, USA) and then subjected to reverse transcription using the PrimeScript RT reagent kit (Takara Bio Inc., Shiga, Japan). Quantitative real-time RT-PCR was performed using a StepOnePlus PCR system (Applied Biosystems, Bedford, MA, USA) with SYBR Premix Ex Taq (Takara Bio Inc., Shiga, Japan). The PCR primer sequences were as follows: human GAPDH, 5′-CAACTACATGGTTTACATGTTC-3′ (forward) and 5′-GCCAGTGGACTCCACGAC-3′ (reverse); human p16, 5′-CGAATAGTTACGGTCGGAGG-3′ (forward) and 5′-TGAGAGTGGCGGGGTCG-3′ (reverse); human p21, 5′-TCAGGGTCGAAAACGGCG-3′ (forward) and 5′-AAGATCAGCCGGCGTTTGGA-3′ (reverse); human Lamin B1, 5′-GGGAAGTTTATTCGCTTGAAGA-3′ (forward) and 5′-ATCTCCCAGCCTCCCATT-3′ (reverse); human IL-6, 5′-CCAGGAGCCCAGCTATGAAC-3′ (forward) and 5′-CCCAGGGAGAAGGCAACTG-3′ (reverse); human RTN4,5′-GGGTGTGATCCAAGCTATCC-3′ (forward) and 5′-CGCCTGAGTTCCTTTATCGT-3′ (reverse); human ATP6V0D1,5′-GGACAATGGCTACTTGGAGGG-3′ (forward) and 5′-GCAGATGCAGTTTCAAGTCCTCTA-3′ (reverse). The means ± s.d. of three independent experiments are shown.

### 4.8. Western Blot Analysis

The cells were lysed using the RIPA buffer (25 mM Tris-HCl pH 7.6, 150 mM NaCl, 1% NP-40, 1% sodium deoxycholate, and 0.1% SDS) supplemented with a 1% Protease inhibitor cocktail (Nacalai Tesque, Kyoto, Japan). The protein concentrations were determined using a Pierce™ BCA Protein Assay Kit (Thermo Fisher Scientific, Waltham, MA, USA, #23225). The proteins were isolated using SDS-PAGE and transferred onto PVDF membranes (Merck Millipore, MA, USA). After blocking the membranes with 5% skim milk (Megmilk Snow Brand Co., Ltd., Hokkaido, Japan) in Tris-buffered saline with 0.1% Tween 20 (TBST), they were incubated with primary antibodies, such as CD9 (Abcam, Cambridge, UK, #ab92726, 1:1000 dilution), CD81 (Cosmo-Bio, Tokyo, Japan, #SHI-EXO-M03, 1:1000 dilution; SBI, #EXOAB-CD81A-1, 1:1000 dilution), CD63 (Cosmo-Bio, Tokyo, Japan, #SHI-EXO-M02, 1:1000 dilution), Nogo-A (Abcam, Cambridge, UK, #ab62024, 1:1000 dilution), ATP6V0D1 (Proteintech, IL, USA, #18274-1-AP, 1:1000 dilution), α-Tubulin (Millipore Sigma, MA, USA, #T9026, 1:10,000 dilution), and calnexin (Cell Signaling Technology, Danvers, MA, USA, #CST2679, 1:1000 dilution) overnight at 4 °C in blocking buffer. The membranes were then incubated with secondary antibodies (GE Healthcare, Chicago, IL, USA, 1:10,000 dilution) for 40 min at room temperature, visualized using the SuperSignal West Pico PLUS Chemiluminescent Substrate or the SuperSignal West Femto Maximum Sensitivity Substrate (Thermo Fisher Scientific, Waltham, MA, USA), and detected using the FUSION SOLO S (Vilber Lourmat, Collegien, France).

### 4.9. Clinical Samples

Human skin fibroblasts NF1, WF1 and WF2 cells were investigated as previously described [27]. NF1 cells were collected from a 42-year-old Japanese healthy male. WF1 cells were collected from a 47-year-old Japanese male who was diagnosed with Werner syndrome (WS), and WF2 cells were collected from a 43-year-old Japanese male who was diagnosed with WS. WF1 and WF2 cells had a homozygous mutation in the WRN gene (Mut4 mutation: c.3139-1G > C). These cells were collected after obtaining informed consent for genetic and cell biological analyses from the patients. The cells were cultured in DMEM supplemented with 10% FBS and 1% penicillin/streptomycin. The study was conducted in accordance with protocols approved by the Institutional Review Board (approval number: 2019-1211) of the JFCR. When sEVs were collected from the culture supernatant of WF cells, we confirmed that cell proliferation had been arrested. Normal fibroblasts (NF) cells, which serve as the control, were passaged according to the population doubling level (PDL) number of WF cells. The sEVs were collected at the same PDL number as WF cells by size exclusion chromatography (SEC).

### 4.10. Statistical Analysis

Statistical analysis was conducted using an unpaired two-tailed Student’s *t*-test in Excel (Microsoft). *p*-values < 0.05 were considered statistically significant. Receiver operating characteristic (ROC) curves were generated to calculate the area under the curve using the pROC package (version 1.18.0) [44]. Transcriptome data of human dermal fibroblasts in people aged 1–94 years old [39] were divided into two categories: young fibroblast (<40, n = 65) or aged fibroblast (>50, n = 62).

## 5. Conclusions

In conclusion, we identified ATP6V0D1 and RTN4 as novel senescent markers of sEVs that were isolated by three different purification methods (SEC, affinity method for PS, and IP) (Figure 5). We confirmed that these proteins were enriched in sEVs from progeroid cells and the serum of aged mice. These findings indicate that sEVs may be useful in detecting the presence of senescent cells in vivo.

## Figures and Tables

**Figure 1 ijms-24-02421-f001:**
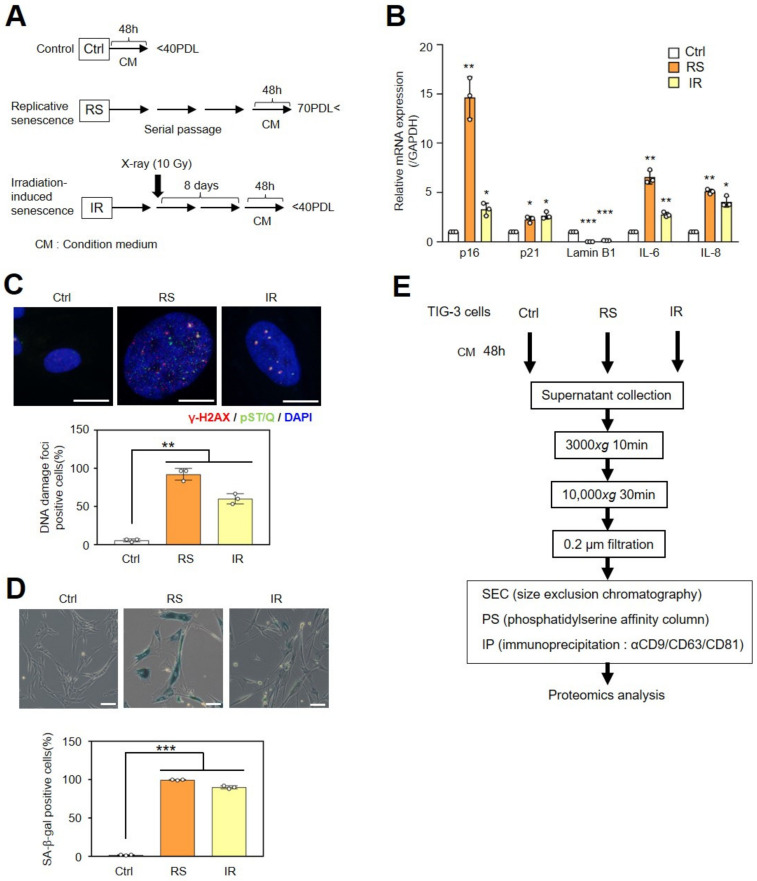
Collection of small EVs using three methods. (**A**) The timeline of senescence induction and medium collection using TIG-3 cells are shown. After induction of cellular senescence, control (Ctrl) and senescent cells (RS and IR) were cultured in conditioned medium (CM) for 48 h. (**B**) Relative mRNA levels of p16^INK4a^, p21^WAF/CIP1^, lamin B1, IL-6, and IL-8 in control and senescent TIG-3 cells were detected using RT-qPCR. Relative quantitation data represent the mean ± standard deviation normalized to GAPDH. (**C**) Immunofluorescence staining with DNA damage markers, γ-H2AX (red), phospho-(Ser/Thr) ATM/ATR (pST/Q) substrate (green), and DAPI (blue). Scale bars = 20 µm. Quantification of DNA damage-positive cells. The histograms indicate the percentage of nuclei containing more than two positive foci for both γ-H2AX and pST/Q staining from at least 100 cells per condition for three independent experiments. (**D**) Senescence-associated β-galactosidase (SA-β-gal) staining. The histograms indicate the percentage of SA-β-gal positive cells. Scale bars = 100 µm. (**E**) The procedure for recovering sEVs from the CM uses three different collection methods. For all graphs, *p*-values were calculated by an unpaired two-tailed Student’s *t*-test (* *p* < 0.05, ** *p* < 0.01, or *** *p* < 0.001 by the unpaired two-sided *t*-test).

**Figure 2 ijms-24-02421-f002:**
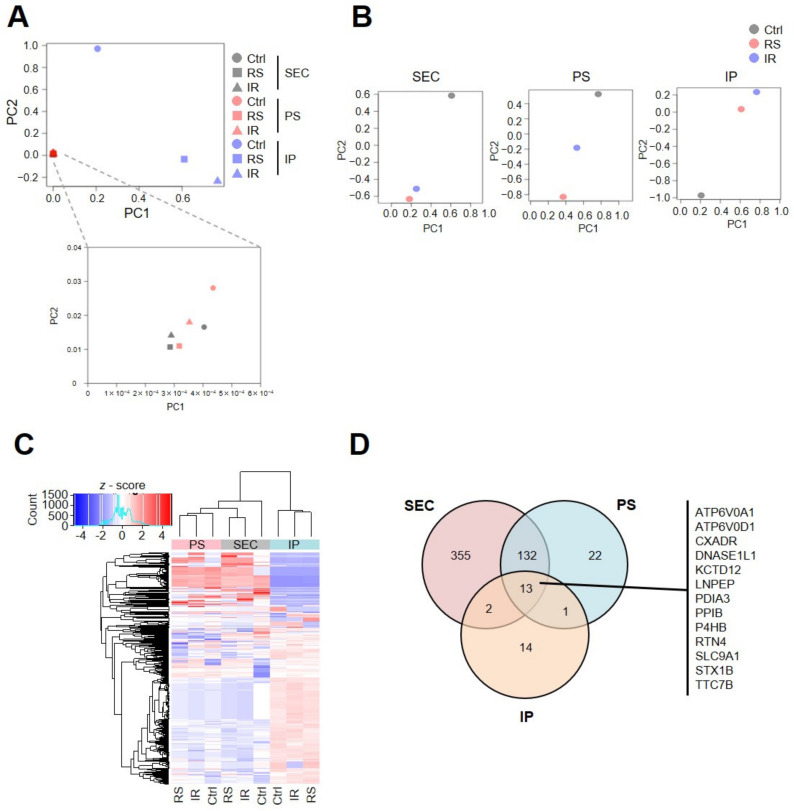
Proteomic characterization of smallEVs. (**A**) Principal component analysis (PCA) performed with the prcomp function using the log10 abundances calculated by mass spectrometry analysis of sEVs in Figure 1E. PCA was performed on all samples (**A**) or each collection method (**B**), as shown. In the lower panel of Figure 2A, the enlarged figure is shown in SEC and PS. (**C**) Heatmap corresponding to proteins detected by mass spectrometry analysis of sEVs in Figure 1E. Each column shows the values normalized to the z-score. (**D**) For each collection method, proteins with an abundance ratio of RS-EVs to Ctrl-EVs and IR-EVs to Ctrl-EVs that were more than twice were extracted. Among them, thirteen proteins were common in sEVs collected by the three methods.

**Figure 3 ijms-24-02421-f003:**
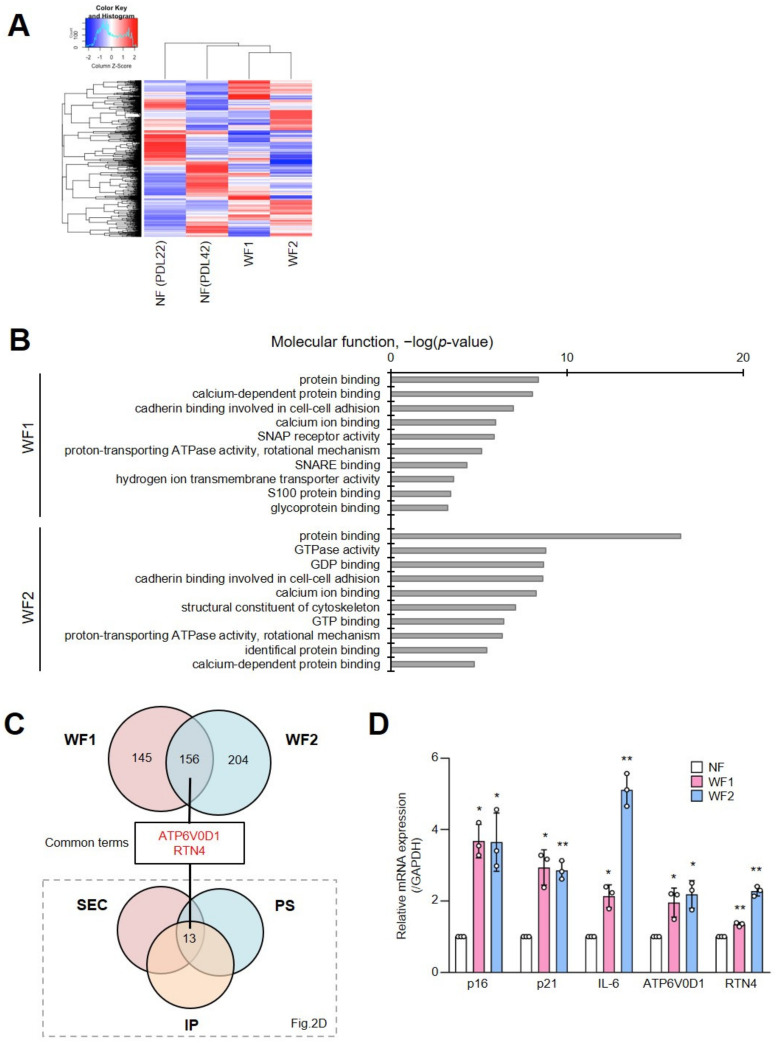
Proteomic characterization of small EVs from the cells of patient with Werner syndrome. (**A**) Cluster analysis was performed using the proteomic data of NF-derived EVs and WF-derived EVs. (**B**) Gene Ontology (GO) analysis of differentially expressed genes was performed using the database for annotation, visualization and integrated discovery (DAVID). The 10 most significantly (** *p* < 0.01) enriched GO terms for molecular function are presented. (**C**) The common terms of proteins whose abundance ratio was more than twice that of the two WF-derived EVs to NF-derived EVs were examined, and 156 proteins were detected. This common term and the common term of thirteen proteins narrowed down in TIG3 cell-derived EVs (Figure 2D) were examined, and only ATP6V0D1 and RTN4 were detected. (**D**) Relative mRNA expression of p16^INK4a^, p21^WAF/CIP1^, IL-6, RTN4, and ATP6V0D1 are shown in the bar chart. Relative quantitation data represent the mean ± standard deviation normalized to GAPDH. Statistical analysis was performed using a two-tailed unpaired Student’s *t*-test (* *p* < 0.05 and ** *p* < 0.01 by the unpaired two-sided *t*-test).

**Figure 4 ijms-24-02421-f004:**
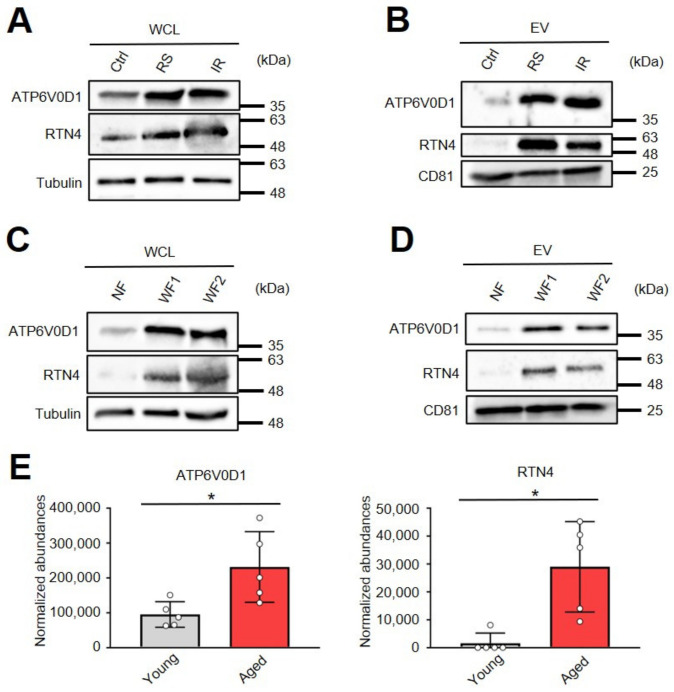
Two senescent markers detected in small EVs. (**A**,**B**) Western blot analysis of whole-cell lysate (WCL) and purified sEVs from TIG-3 cells. (**C**,**D**) Western blot analysis of WCL and purified sEVs from normal fibroblasts (NF) and Werner syndrome patient-derived fibroblasts (WF). (**E**) The normalized abundance of ATP6V0D1 and RTN4 protein in serum-derived EVs of 8-week-young mice (n = 5) and 110-week-aged mice (n = 5). Statistical analysis was performed using a two-tailed unpaired Student’s *t*-test (* *p* < 0.05 by the unpaired two-sided *t*-test).

**Figure 5 ijms-24-02421-f005:**
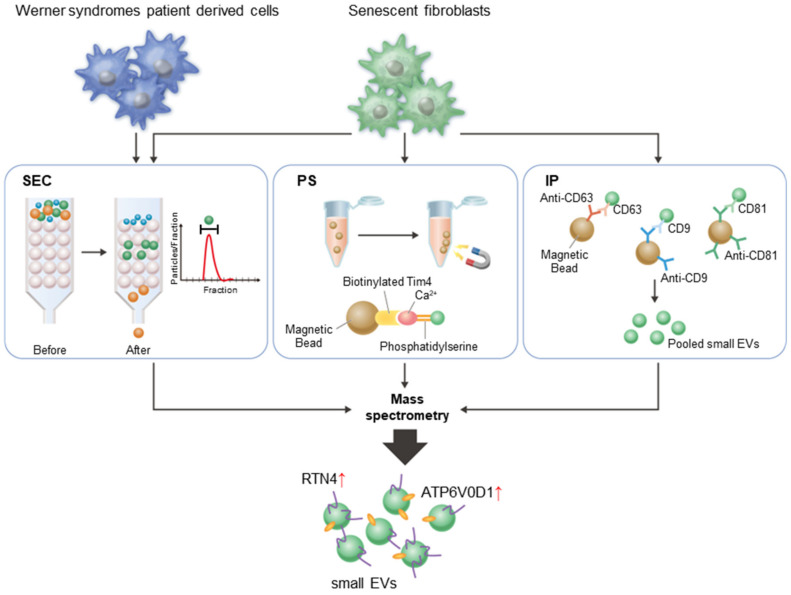
Graphic abstract for this study. sEVs were purified from culture supernatants of senescent fibroblasts using three different purification methods and analyzed by mass spectrometry. sEVs were also purified from culture supernatants of cells derived from patients with Werner syndrome using SEC and analyzed by mass spectrometry. The results of the two mass spectrometry analyses identified ATP6V0D1 and RTN4 as cellular senescence markers enriched in sEVs. SEC: size exclusion chromatography, PS: affinity column for phosphatidylserine, IP: immunoprecipitation with anti-CD9, -CD81, and-CD63 antibodies. Red arrows indicate that ATP6V0D1 and RTN4 are highly expressed in sEVs derived from senescent cells.

## Data Availability

The data that support the findings of this study are available in the Appendix A of this article. The raw data utilized in this study will be made publicly available upon publication via Japan Proteome Standard Repository/Database (jPOST), ID: JPST001902.

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
