# Peer review of "Identification of Novel Senescent Markers in Small Extracellular Vesicles"

_ijms, 2023, doi:10.3390/ijms24032421_

Round 1
Reviewer 1 Report
Misawa et al. analyze provide a detailed combinatorial analysis of small EV isolation using different methods from senescent cells, induced by different stressors. Proteomic analysis shows the overlap and distinctive signatures in these small EVs. Further analysis is done using EVs derived from Werner's Syndrome patient cell lines. The authors identify ATP6V0D1 and RTN1 as senescence-related markers found in EVs. These two sEV markers were found to be enriched in serum EVs isolated from naturally aged mice. This work presents a use for these two markers as readout for both tissue and systemic senescent cell burden. The manuscript is well-written and logically ordered. The experiments are properly designed and controlled. I think this is a fantastic manuscript and I enthusiastically support it being published, therefore I recommend acceptance as is of this manuscript.
Author Response
Thank you. We are grateful with this reviewer’s comment.
Reviewer 2 Report
This is a very well written, very clear and well designed study, aiming at identifying novel senescent markers in sEV. The authors propose ATP6V0D1 and RTN4 as novel senescent markers, after the demonstration that these 2 proteins are up-regulated in sEV from cultured senescent cells and in sEV from cells derived from patients with Werner Syndrome, and that these proteins are over-expressed in sEV from the serum of aged mice.
My only comment concerns the potential utility of these 2 proteins in vivo: with the exception of the serum from aged mice, this study was performed in vitro using cultured cells, not in vivo. The Discussion should therefore address this limitation; the authors propose that ATP6V0D1 and RTN4 could be detected in body fluids such as blood, urine or saliva, but only blood was tested here.
In addition, the advantages of these novel sEV markers over SASP or other classical senescent markers should be better presented in the Discussion: some SASP markers can also be detected in the serum/plasma, with simple and rapid ELISA test, so why using ATP6V0D1 and RTN4 from sEV?
Author Response
This is a very well written, very clear and well designed study, aiming at identifying novel senescent markers in sEV. The authors propose ATP6V0D1 and RTN4 as novel senescent markers, after the demonstration that these 2 proteins are up-regulated in sEV from cultured senescent cells and in sEV from cells derived from patients with Werner Syndrome, and that these proteins are over-expressed in sEV from the serum of aged mice.
Response-1:
Thank you very much for reviewing our manuscript and offering valuable advice.
My only comment concerns the potential utility of these 2 proteins in vivo: with the exception of the serum from aged mice, this study was performed in vitro using cultured cells, not in vivo. The Discussion should therefore address this limitation; the authors propose that ATP6V0D1 and RTN4 could be detected in body fluids such as blood, urine or saliva, but only blood was tested here.
Response-2:
We agree with the reviewer. As you pointed out, we only confirmed the presence of ATP6V0D1 and RTN4 in blood samples in the in vivo experiments of this study. In line with the reviewer’s suggestion, we have revised the sentence in discussion in the main text as described page 8, lines 257 to 258.
In addition, the advantages of these novel sEV markers over SASP or other classical senescent markers should be better presented in the Discussion: some SASP markers can also be detected in the serum/plasma, with simple and rapid ELISA test, so why using ATP6V0D1 and RTN4 from sEV?
Response-3:
You have raised an important point. We think that it is difficult to evaluate cellular senescence by SASP factor alone because SASP factor is upregulated not only by cellular senescence but also by inflammation and other factors (Faget et al., Nat. Rev. Cancer, 2019). It is eventually expected that SASP factors will be detectable from a blood sample. However, since blood contains albumin, many free proteins and protein aggregates, methods of detecting these proteins in blood components may not be sensitive enough. In contrast, proteins within sEVs are stable because they possess a lipid bilayer structure. Therefore, we believe that proteins enriched in sEVs (ATP6V0D1 and RTN4) may increase the sensitivity and accuracy of the detection of senescent cell from blood samples. We have added these points in discussion. Please check the first paragraph on page 8, lines 221 to 243.